# Non-Contact Breathing Monitoring Using Sleep Breathing Detection Algorithm (SBDA) Based on UWB Radar Sensors

**DOI:** 10.3390/s22145249

**Published:** 2022-07-13

**Authors:** Muhammad Husaini, Latifah Munirah Kamarudin, Ammar Zakaria, Intan Kartika Kamarudin, Muhammad Amin Ibrahim, Hiromitsu Nishizaki, Masahiro Toyoura, Xiaoyang Mao

**Affiliations:** 1Faculty of Electronic Engineering Technology, Universiti Malaysia Perlis (UniMAP), Arau 02600, Perlis, Malaysia; husaininadri@studentmail.unimap.edu.my (M.H.); ammarzakaria@unimap.edu.my (A.Z.); 2Advanced Sensor Technology, Centre of Exellence (CEASTech), Universiti Malaysia Perlis (UniMAP), Arau 02600, Perlis, Malaysia; 3Centre of Advanced Sensor and Technology, Universiti Malaysia Perlis, Arau 02600, Perlis, Malaysia; 4Department of Otorhinolaryngology Head and Neck Surgery, Universiti Teknologi MARA, Shah Alam 47000, Selangor, Malaysia; kartika@uitm.edu.my; 5Department of Internal Medicine, Faculty of Medicine, Universiti Teknologi MARA, Shah Alam 47000, Selangor, Malaysia; m_amin88@uitm.edu.my; 6Faculty of Engineering, University of Yamanashi, Kofu 400-8511, Yamanashi, Japan; hnishi@yamanashi.ac.jp; 7Department of Computer Science and Engineering, University of Yamanashi, Kofu 400-8511, Yamanashi, Japan; mtoyoura@yamanashi.ac.jp (M.T.); mao@yamanashi.ac.jp (X.M.)

**Keywords:** breathing rate (BR), sleeping monitoring, contactless sensing, polysomnography (PSG), ultra-wideband (UWB) radar

## Abstract

Ultra-wideband radar application for sleep breathing monitoring is hampered by the difficulty of obtaining breathing signals for non-stationary subjects. This occurs due to imprecise signal clutter removal and poor body movement removal algorithms for extracting accurate breathing signals. Therefore, this paper proposed a Sleep Breathing Detection Algorithm (SBDA) to address this challenge. First, SBDA introduces the combination of variance feature with Discrete Wavelet Transform (DWT) to tackle the issue of clutter signals. This method used Daubechies wavelets with five levels of decomposition to satisfy the signal-to-noise ratio in the signal. Second, SBDA implements a curve fit based sinusoidal pattern algorithm for detecting periodic motion. The measurement was taken by comparing the R-square value to differentiate between chest and body movements. Last but not least, SBDA applied the Ensemble Empirical Mode Decomposition (EEMD) method for extracting breathing signals before transforming the signal to the frequency domain using Fast Fourier Transform (FFT) to obtain breathing rate. The analysis was conducted on 15 subjects with normal and abnormal ratings for sleep monitoring. All results were compared with two existing methods obtained from previous literature with Polysomnography (PSG) devices. The result found that SBDA effectively monitors breathing using IR-UWB as it has the lowest average percentage error with only 6.12% compared to the other two existing methods from past research implemented in this dataset.

## 1. Introduction

In healthcare, Breathing Rate (BR) is one of the most meaningful indicators for inspecting human vital signs. Almost all the time, BR conveys the information about clinical deterioration, analyses severe pneumonia stages, and predicts cardiac arrest. In addition, the state of BR expresses an outcome in advance for admission into the Intensive Care Unit (ICU) [1] and hospital mortality [2]. In general, normal BR (eupnea) for a healthy adult would range from 12 breaths/min to 20 breaths/min [3], while for children, the normal value varies corresponding on their age [4]. Typically, abnormal BR can be separated into three main classes, namely Bradypnea (patient with low breathing rate), Tachypnea (patient with high breathing rate), and Apnea (cessation of breathing). Apnea can be branched into two main groups called Obstructive Apnea, resulting from airway obstruction and Central Apnea, because of respiratory system development deficiencies [5]. Yuan et al. [3] added a few more abnormal BR such as Biot’s breathing, Ataxic breathing, Kussmaul’s breathing, Thoracoabdominal paradox, Cheyne–Stokes respiration, Apneustic breathing, and Agonal breathing.

For low-acuity patients in a nursing home or general hospital, the majority of the time, BR was taken by manual measurement. A doctor or medical practitioner visually counted the patient breath in a sixty-second window. In most cases, manual counting exposes to errors during assessments [6] as these medical practitioners either bias count to 18 or 20 breaths per minute (BPM) for each patient [7]. The alternative BR monitoring such as pulse oximetry, breathing belt, electrocardiogram (ECG), and photoplethysmography (PPG) may be suitable for frequent monitoring of patients’ BR in intensive care because, in many cases, nurses missed 90% of smoothed hypoxemic episodes due to relying on manual intermittent spot check [8]. These contact-based sensors technology provide good accuracy in measuring BR [9] and have been used in many hospitals. However, these sensors are not feasible for long-term monitoring as they may cause discomfort, disconnection due to cable kinks, and epidermal stripping, such as in infants monitoring, patients with severe skin burns, or sleep monitoring.

In relation to these concerns, non-contact sensors such as optical and sonar systems have shown up in many different types of applications including in healthcare. For optical systems, computer vision is generally implemented in the system where the latter technology used Remote Photoplethysmography (remote-PPG) with a low-cost camera [10]. There are two types of cameras used for breathing monitoring: thermal and RGB. The camera sensor extracts RR by measuring the intensity of light reflected by skin color. The main challenge of this sensor is limited to the ambient light levels where the low-level light might distort the respiratory signal. Another option is to use a sonar system. A Sonar system is a method that analyses the acoustic signal obtained from a microphone during inhalation and exhalation processes [11]. Ge et al. [12] have proved the capability of acoustic signals in estimating RR by analyzing the cheap built-in microphone inside the smartphone under various scenarios. However, there are some limitations to this approach, for example, the distance of the patient’s chest to the microphone plays a vital role in the experiment. When the distance becomes larger, the error gets larger. Furthermore, acoustic signals are sensitive to environmental noise or other body sounds, such as speaking, sneezing, and heartbeats [13].

Another non-contact measuring technique is radar. The radar sensor works by transmitting the echo signals from the transmitter to detect the chest displacement induced by the respiration process and analyze the changes form of the received signal for contactless RR measurement. Radar technology has the potential to provide contactless, continuous, and long-term RR monitoring. Additionally, radar also has the potential to be a simple and innovative portable monitoring solution for use at home. It is less invasive and handier than current portable devices that use contact sensors since it does not require contact with patients [14]. Furthermore, radars such as FMCW have spatial resolution ideal for such a short-range application [15].

In relation to these concerns, non-contact sensors such as UWB radar have shown up in many different types of applications, including in healthcare. UWB radar is a short-range wireless communication technology that uses periodic pulse signals from the transmitter, and unlicensed operation is permitted under new restrictions to operate in the frequency range of 3.1–10.20 GHz [16]. As a result, compared to other radars, such as continuous wave doppler radar, UWB radar provided larger detection areas of up to 10 m. UWB radar has been utilized in a wide range of applications, including people counting [17], in-car mobile phone usage detection [18], hand pointing gesture detection [19], and obstructive sleep apnea detection [20]. Wang et al. (2020) [21], in their research, have made a fair comparison regarding the performance of both Impulse Radio Ultra-Wideband (IR-UWB) and Frequency Modulation Continuous Wave (FMCW) radars in measuring noncontact vital signs such as respiration rate and heart rate. Their overall result found that the UWB radar demonstrated better accuracy ratios and higher SNR than the FMCW radar in all comparative scenarios. Even though UWB has numerous advantages, it also has significant drawbacks. UWB technology may interfere both GPS and aircraft navigation radio equipment, as well as current systems operating in the ultra-wide spectrum [22]. Third generation (3G) wireless technologies, for example, may be affected in some regions [23]. UWB operates by sending out a wideband bandwidth of short pulse waves every millisecond, offering a high spatial resolution of micro-movement from the human body to be obtained. As a result, the received pulse generates amplitude and phase variation that may be used for estimating BR.

Limb movements are quite common for some people during sleep. Report [24] shows that average sleepers tend to make 40 to 50 times movement per night, and the number changes depending on a certain situation. Therefore, it would be unrealistic to expect subjects that participate to remain static for a long period of time during sleep. Thus, a robust random body motion detection technique is required to remove the random body movement. The breathing pattern caused by chest wall movement can be effortlessly recognized in a stationary state condition, considering the subject body posture and chest location remains at the same cell range of UWB radar. Meanwhile, any macro body movement (e.g., shivering or position shift) during sleep might cause a substantial disruption of a signal at the same cell range in UWB radar, making it difficult to extract breathing signals. This happens because chest location will fall into different cell ranges in UWB compared during the stationary state. As a result, the traditional signal processing approach may find it difficult to extract breathing signals. So, it is necessary to cut out any noise produced by body motion for better signal extraction. Therefore, this paper proposed an SBDA algorithm for extracting BR for static and body movement detection during sleep.

This paper proposed a Sleep Breathing Detection Algorithm (SBDA) for extracting the BR of human subjects during sleep using UWB radar. The main contributions of the paper are as follows:This paper discusses the implementation of a variance feature with wavelet decomposition for signal clutter removal using an interval dependent threshold method.Periodic motion detection was detected by monitoring the periodicity of the signal via applying a sine curve fitting algorithm to match the output signal through a calculated R-square value.Ensemble Empirical Mode Decomposition (EEMD) with a Fast Fourier Transform (FFT) was implemented for extracting BR.

## 2. Related Work

UWB radar has gained popularity as a substantial amount of study has been conducted on non-invasive BR monitoring using UWB radar in the last few years due to advantages in range resolution as well as low power consumption. Although many impact works of literature have been published in recent years, selecting reliable signal processing algorithms is still an open research area [25]. Compared with various signal processing techniques, Fast Fourier Transform (FFT) is one of the most popular and basic methods for extracting BR. Many researchers, for example, Refs. [26,27], implement FFT in their research on extracting BR. FFT is able to estimate BR by obtaining the peak of the frequency spectrum attained from the FFT transformation of time-domain signals at a specific frequency range. At the same time, FFT has an issue of poor spectral resolution. In some cases, FFT, by mistake extracts breathing harmonic as BR [28]. To resolve the problem, Ref. [29] implement Permutation Entropy (PE) and Empirical Mode Decomposition (EMD) to denoise the signal before applying FFT for BR estimation. EMD, however very sensitive to noise, especially for the non-stationary subject.

Detection of BR becomes problematic when body movement is involved during data collection. Many approaches have been developed to enhance BR detection to remove significant body movement artefacts. Khan and Cho [30] addressed the issue of measuring non-stationary human BR. They implement a Kalman filter to reduce clutter signals before applying an auto-correlation algorithm to detect body motion during the experimentation process. However, the approach is very simple as the measurement of BR is only taken when the subject is stationary, while for non-stationary conditions, the value is discarded. Lazaro et al. [31] proposed the body movement removal technique by applying a threshold method from moving average filter after clutter suppression stage. The threshold value was set to 2.5 since human chest displacement is smaller than 2.5 cm. However, this method also discarded the signal if body motion was detected. Adib et al. [32] applied a window and FFT for detecting random body movement. They segment each signal into 30 s window to measure each signal’s periodicity in the window. The periodicity was measured by comparing FFT’s peak frequency amplitude with their threshold. This technique also discards the signal when movement is detected, similar to the other methods. In recent times, Zhang et al. [33] have introduced a feasible method for monitoring BR while moving. They address this challenge by placing two UWB radars at two different locations: the chest and abdomen of the subject for collecting data simultaneously. The body movement was alleviated by implementing Empirical Wavelet Transform (EWT) before applying a cross-correlation function for chest and abdomen movement signals to obtain accurate BR. This method, however, required extra cost since they used more than one UWB radars in the research.

## 3. Method

### 3.1. Data Collection

The experimental design is shown in Figure 1. It comprises a UWB radar supported by a Polyvinyl Chloride (PVC) bar having an approximate range of 0.5 to 2 m from the human chest to prevent the radar from moving or shaking. In this study, Novelda’s IR-UWB radar chip X4M200 center frequency of 7.29 GHz and a bandwidth of 1.5 GHz was utilized to record the human respiration pattern, and an Acer Aspire laptop was used to monitor the radar chip during the data collection. The radar was digitized at a rate of 20 frames per second, with a detection zone of around 3 m. The UWB radar was attached to the laptop via Universal Asynchronous Receiver-Transmitter (UART), whereas MATLAB software was employed to collect data and analyze the signal for BR and HR extraction. During the experiment, the subject lay on the bed and slept in his preferred position while wearing the PSG device for data validation. The Polysomnography data was recorded using Noxturnal (Nox Medical) software. Thoracic and abdominal respiratory effort, blood oxygen saturation, body posture, and nasal airflow were measured. The range of sleep duration for all subjects is between 4 and 8 h.

There were 15 adult subjects recruited among Universiti Malaysia Perlis (Unimap) students, staff, family members, and local communities who live in Perlis, Malaysia. Table 1 shows the detail of the subjects. The subjects included nine females and six males ranging in age from 21 to 42 years. Furthermore, 9 subjects were classified with a normal rating and 6 subjects were diagnosed with an abnormal rating. The potential subjects will be rejected if they have one of the following criteria: (1) positive COVID-19, or (2) close contact with the COVID-19 patients. Information sheets and consent forms will be given after subjects agree to participate in the study. The selected subjects must fill out the STOPBANG Questionnaire, Epworth Sleepiness Scale, and Subject proforma form via a given google form link. This research adopted four STOPBANG questionnaire questions and four extra demographic questions for STOPBANG Questionnaire. The total of eight dichotomous (yes/no) questions included age, gender, tiredness, neck circumference, snoring, BMI, high blood pressure, and observed apnea score. Patients with a STOPBANG score of 0 to 2 were classified as low risk, 3 to 4 classified as midrange, and 5 to 8 classified as high risk for obstructive sleep apnea [34]. This study also applied Epworth Sleepiness Scale for measuring daytime sleepiness. The Epworth Sleepiness Scale has eight self-rated items, ranging from 0 to 3. A score of more than ten suggests excessive daytime drowsiness [35]. An appointment for data collection is scheduled with the subject before performing data collection at the subject’s place. Two days before the appointment date, all subjects must fill out the health declaration form to ensure the subject is free from COVID-19 or other communicable diseases. The abnormalities rating for each patient was based on an AHI score of more than 5, an Epworth Sleepiness Scale score of more than 10, and a STOPBANG score of more than 5.

### 3.2. UWB Radar Working Principle

When employing radar to monitor vital signs, the standard procedure involves monitoring the thoracic displacement induced by actions such as respiration and heartbeat. Generally, UWB radar determines the vital signals by measuring the amplitude of an estimated human body point. The UWB radar works by transmitting the pulses and hitting the human body. Thoracic motion changes the frequency and phase of a signal, causing it to be demodulated. The front surface of the chest is the primary source of reflection signals, which are used in the respiration detection process. The fluctuations in the reflection signal, including the pulse width and pulse amplitude, are caused by the changes in the distance between the chest surfaces that occur during inhale and exhale. Thus, the received signal contains vital signs information.

For monitoring human breathing, this study adopted a UWB radar system on chip (SoC) X4M200 in the experiment. The X4M200 UWB sensor is primarily a RADAR-based sensor with integrated transmitter and receiver antennas into a SoC design. This type of radar has the capability of producing highly accurate measurements of respiration as well as a respiratory detecting range detection that may extend up to 5 m [36]. UWB radars such as the XeThru X4M200 radar are simple to install, affordable, have minimal multipath propagation, and can even get through solid walls.

Figure 2 reveals the X4M200 UWB radar sensor, and Figure 3 give a complete block schematic of the device.

### 3.3. Sleep Breathing Detection Algorithm (SBDA)

This paper proposed a Sleep Breathing Detection Algorithm (SBDA). This study recruited fifteen volunteers with a normal and abnormal ratings from students and staff of Universiti Malaysia Perlis, friends, and local communities in Perlis Malaysia.

#### 3.3.1. Clutter Suppression

During the experiment, the subject lay on the bed and slept in his preferred position while wearing the PSG device for data validation. Continuous pulses were sent from the radar transmitter. The receiver would receive the signals transmitted by the subject. The signals acquired were stored in a matrix form of size (*m* × *n*), where m and n denote the number of samples in slow and fast time, respectively. The received waveforms are measured periodically in slow time. *N* discrete-time sequences are stored after the received signal is sampled. These values are stored in a matrix $R_{M\times N}$, the elements of which are:(1)$$R_{M\times N}=R\;\left(\tau =mT_{f},\;t=nT_{s}\right)$$
where $\tau =mT_{f}$ (*m* = 1, 2,…, *M*) $T_{f}$ is the sampling period in fast time, and $t=nT_{s}$ (*n* = 1, 2, …, *N*) $T_{s}$ is the sampling interval in slow time.

The signal obtained includes clutters generated from the surroundings and vital sign signals from the subject’s chest. In a static environment, the resulting clutter represents a DC component in the slow-time direction. In such an environment, the only movement is caused by the person’s breathing and heart activity, and the background clutter does not depend on slow time. According to Antonio Lazaro et al. [31], the modeling of static clutter can be represented by:(2)$$C\left(t,\tau \right)=\alpha \cdot C\left(t-1,\tau \right)+\left(1-\alpha \right)\cdot x\left(t,\tau \right)$$
where *t* is the slow-time, *τ* is the fast-time delay, $C\left(t,\tau \right)$ is the current clutter, $x\left(t,\tau \right)$ is the raw echo signal, $C\left(t-1,\tau \right)$ is a previous clutter, and α is the suppressing parameter between [0, 1]. If α = 1, the current clutter is equal to the previous clutter. If α = 0, the current clutter is equal to the raw echo signal.

The first step was to choose the range in which the subject is found. There is minimal movement because there is only one subject in the detection range, and the subject is sleeping. The breathing movement will produce the highest energy within the detection range. The location can be found by selecting the value of variance across slow-time UWB radar, which contains a respiration signal. The width of the variance curve was preserved for further denoise processing.

The next step is to implement wavelet transform. The wavelet transform is one of the powerful techniques for assessing non-stationary signals due to its capabilities to extract time-frequency features from the signal compared to other transforms. There are two well-known types of wavelet transforms, which are Discrete Wavelet Transform (DWT) and Continuous Wavelet Transform (CWT) [37]. The significant contrast among CWT and DWT is how the scale parameter is discretized. Compared to the DWT, the CWT discretizes scale more finely since scaling and shifting for CWT can have all possible values. In contrast, DWT is always discretized to integer powers of 2, so that the number of voices per octave is always 1 [38]. Thus, DWT coefficients are only produced at certain scales and time periods [39]. Therefore, in this paper, Discrete Wavelet Transform (DWT) is used instead of the CWT since it is more suitable for signal de-noising as it does not require a long computation time, as well as a huge amount of workload, is reduced.

For the DWT computation, the scale parameter and translation parameter can be represented in Equations (3) and (4):(3)$$a=2^{j}$$
(4)$$\tau =ka=k2^{j}$$
where a is the dilation factor, τ is the translation factor, j is the scale index, and k is the wavelet transform signal index.

Therefore, DWT detail coefficients at level *j* can be expressed as follows:(5)$$D_{j}=\overset{\infty }{\underset{n=0}{\sum }}f\left(n\right)\frac{1}{\sqrt{2^{j}}}\;\psi \;\left(\frac{n-k2^{j}}{2^{j}}\right)$$

The general wavelet de-noising procedure in this paper involves three steps as detailed below:A decompose the signal in order to satisfy the signal-to-noise ratio in the signal.Select threshold detail coefficients. This paper adopts a dependent threshold method due to the behavior of the signal. The threshold was selected based on the width of the variance curve signal. A soft threshold was applied to the detail coefficients.Reconstruction. To reconstruct the signal, Inverse Wavelet Transform (IWT) was used to obtain a de-noised signal.

#### 3.3.2. Periodic Motion Detection

Some people have frequent movement patterns while sleeping. According to [15], average sleepers move 40 to 50 times per night, with the number varying depending on the situation. As a result, expecting subjects in this study to remain still for an extended amount of time during sleep would be unrealistic. So, our objective in this section is to find the parts of the signals which contain the finest vital signs information and are least distorted by the body movement and orientation. For that reason, sinusoidal fitting technique was used to discover which signals had nicer sinusoidal forms, and hence revealed better vital sign information. The slow-time sample for each data matrix was analyzed to determine which samples match the sinusoidal motion induced by breathing. As the chest wall movement operates in periodic motion, the sine curve fitting algorithm will analyze how much of the signal falls into sinusoidal patterns. Figure 4 and Figure 5 show the signal sample with sine curve fit above and below the threshold value. The general form of the algorithm can be represented by:(6)$$y=\overset{n}{\underset{i=1}{\sum }}\alpha_{i}\sin(b_{i}x+c_{i})$$
where α is the amplitude, b is the frequency, c is the phase constant for each sine wave term, and *n* is the number of terms in the series.

The *R*-squared value is used for finding the fit of the signal, which is defined as follows:(7)$$R^{2}=1-\frac{\sum_{i=1}^{n}{\left(y_{i}-{\hat{y}}_{i}\right)}^{2}}{\sum_{i=1}^{n}{\left(y_{i}-{\bar{y}}_{i}\right)}^{2}}$$
where $y_{i}$ is the actual value of the radar signal, ${\hat{y}}_{i}\;$represent the estimated values from the sinusoidal fitting algorithm, whereas ${\bar{y}}_{i}$ shows the mean of y.

The *R*-square values range from zero to one. A value approaching one indicates the signal has a higher sinusoidal pattern, which may contain a breathing signal. On the other hand, a lower *R*-squared value suggests the signal contains noise. The threshold value for the algorithm was set to 0.5.
The general sine curve fitting algorithm procedure in this paper involves three steps as detailed below.Apply the sine curve fit in Equation (6) which gives amplitude, frequency, and phase constant for the best fit to the raw signals.Determine the *R*-square value for each signal using Equation (7).Find the signals with an *R*-square value above a threshold value.

#### 3.3.3. Ensemble Empirical Mode Decomposition Approach

The Ensemble Empirical Mode Decomposition (EEMD) is a technique that has been proposed for solving the mode mixing problem that exists in Empirical Mode Decomposition (EMD) [40]. Generally, EEMD is often used as a decomposing tool for handling the noises and target signals of non-static and non-linear systems [41]. Especially, using EEMD, the breathing signal can be divided from the physiological signals detected through the UWB radar system [42].

An EMD algorithm can adaptively decompose the signal into a series of intrinsic mode functions (IMF) and residual signals according to the characteristics of the signal:(8)$$x\left(t\right)=\overset{n}{\underset{i=1}{\sum }}IMF_{i}\left(t\right)+r_{n}\;\left(t\right)$$
where $IMF_{i}$, i=1…, n denote the number of IMF and $r_{n}$ as residue. The physical meaning of each IMF corresponds to the oscillation characteristics of different scales in the original signal. The number of extrema in each IMF is decreased as the IMF order increases, and the corresponding spectral supports are decreased accordingly [43]. Each IMF is estimated with an iterative process called sifting [44]. An IMF must satisfy two conditions: (i) the number of zero-crossings and extreme values must be equal or differ at most by one; (ii) at any point, the mean value of the upper and lower envelopes is zero.

The EMD method has a disadvantage when oscillations with the same time scale are stored in several IMF or vice versa in one IMF storing oscillations with different time scales. This results in the ineffectiveness of the EMD method due to the effect of mixing modes. Wu and Huang [40] proposed an EEMD algorithm to solve suppress mode mixing issues in the EMD algorithm. EEMD algorithm is able to scale better the ‘true’ IMF component generated by EMD via an ensemble of trials with the addition of white noise to the signal. The details of the EEMD algorithm are as follows:1.Add Gaussian white noise to the input signal $x\left(t\right)$.
(9)$$x_{i}\left(t\right)=x\left(t\right)+u_{i}\left(t\right)$$
where $u_{i}\left(t\right)$ (i=1…m,) are different realizations of Gaussian white noise and $x_{i}\left(t\right)$ indicates the signal with Gaussian white noise.2.Applied EMD algorithm to the signal $x_{i}\left(t\right)$ for obtaining the IMF component.3.Add other white noise to the observation signal again and repeat step 1 and 2 until m times.4.Perform average operation on IMF component which has m times trials for obtaining new $\tilde{IMF\;}$ component.5.Final EEMD can be written as
(10)$$x\left(t\right)=\overset{n}{\underset{k=1}{\sum }}\tilde{IMF_{k}\;}+r_{n}\left(t\right)$$

#### 3.3.4. IMF’s Selection and Signal Reconstruction Methods

This section clarifies the specifications of the developed method for signals reconstruction that requires high precision in the selection process of IMFs. Fourier transforms is performed on each of the IMFs, thereby computing the entire energy of each IMF in frequency domain $E\left(j\right)$, the energy of the respiration signal frequency range (0.2–0.8 Hz) $E_{r}\left(j\right)$. The threshold value of respiration is calculated for each IMF in the frequency domain calculated as follows:(11)$$\frac{E_{r}\left(j\right)}{E\left(j\right)}>\delta_{r}$$

$E\left(j\right)$ is the frequency domain energy of the jth IMF and $E_{r}\left(j\right)$ is the energy in the breathing frequency band of the jth IMF. $\delta_{r}$ is the energy threshold value. In this paper, the energy threshold value $\delta_{r}$ was set to 0.5.

## 4. Result

### 4.1. Clutter Removal

Figure 6 shows the comparison of signal before and after clutter removal. The blue waveform represents the raw signal, whereas the red waveform represents the signal after undergoing a clutter removal process. Figure 7 illustrates the signal with strong static clutter, whereas Figure 8 shows the signal after removing the clutter. The remaining part is the subject’s body location which contains the BR signal.

### 4.2. Respiration Signal Extraction

Table 2 shows the sample of the calculated energy ratio between the energy of each IMF in frequency domain $E\left(j\right)$ and the energy of the respiration signal $E_{r}\left(j\right)$ for one minute of signal breathing extraction for fifteen subjects. It can be seen that the energy ratio is only calculated for IMF 4, IMF 5, IMF 6, and IMF 7 since these components contain frequency within the respiration signal frequency range (0.1–0.35 Hz). This leaves the IMF value of IMF 1, IMF 2, IMF 3, IMF 8, IMF 9, and IMF 10 uncalculated. The threshold value was set to 0.5 for the selection criteria. Therefore, the IMF component with an energy ratio of equal or greater than 0.5 will be selected for breathing signals.

### 4.3. Agreement of Proposed Method with PSG Device

Simultaneous readings from the UWB radar and PSG sensor for fifteen subjects (six males and nine females). The breathing rates were calculated using the extracted signals by different separation methods. The bias of the measured rates between the physiological signals recorded from the PSG sensors and the extracted breathing signal from UWB radar were presented in the Bland–Altman analysis. The horizontal axis corresponded to the average breathing rate between UWB radar and PSG sensor. The vertical axis corresponds to the difference in breathing rate for both UWB and PSG sensors. Figure 9 shows the Bland Altman Plot for SBDA with PSG data for all 15 subjects.

### 4.4. Comparison with Other Methods

In this paper, two previous methods from [45,46] had been compared with SBDA for validation purposes. Shen et al. [45] implement an algorithm for extracting breathing rate by combining an autocorrelation concept to determine subject localization before implementing a fast Fourier transform. Pittella et al. [46], on the other hand, applied several methods for static echo removal from UWB radar. After removing the clutter signal, FFT had been performed in a slow time signal for breathing rate extraction. It was found that the Mean Subtraction method, in combination with FFT, showed fast processing with good accuracy compared to others. Therefore, these two clever methods have been implemented in this experiment for side-by-side performance comparison with the proposed SBDA method.

Table 3 shows the percentage error between SBDA with two previous methods from [45,46]. As seen in Table 2, SBDA recorded the lowest average percentage error compared to others, with only 6.12%, followed by Autocorrelation + FFT with 10.68% and Mean Subtraction + FFT with 15.23%. The overall result also recorded that SBDA has the lowest percentage error for all 15 subjects.

## 5. Conclusions

This paper presents the SBDA approach for detecting breathing during sleep time using IR-UWB radar. SBDA comprises several combinations of well-known methods for solving problems such as signal clutter, periodic movement detection, and breathing signal extraction. The received IR-UWB signal clutter was removed by applying variance and DWT based interval dependent threshold method to denoise the signal. The existing method such as averaging still preserves most of the unwanted clutter in the signal compared to DWT. The DWT works well due to its ability to decompose the signal into several layers to satisfy the signal-to-noise ratio. After that, sine curve fitting is introduced for removing random movement from the subject during data collection. This approach works well in distinguishing the random body movement signal and breathing signal. Last but not least, EEMD is used to extract the true breathing signal before implementing FFT to calculate the rate per minute. The proposed approach is evaluated with two existing methods available in the works of literature in comparison with PSG data. The experimental results reveal the effectiveness of the proposed method, which recorded the lowest average percentage error compared with the other methods. This method also works well for figuring out the rate of breathing in both normal and abnormal patients. Last but not least, this methodology successfully detects breathing rate independently in the presence of body movement and orientation.

This work has some limitations. For data collection, this study only considers 15 subjects. Therefore, more subjects with a variety of breathing disorder track records will be recruited in the future. Furthermore, this work does not address the radar self-motion (RSM) cancellation issue, which may somewhat influence breathing signal extraction.

## Figures and Tables

**Figure 1 sensors-22-05249-f001:**
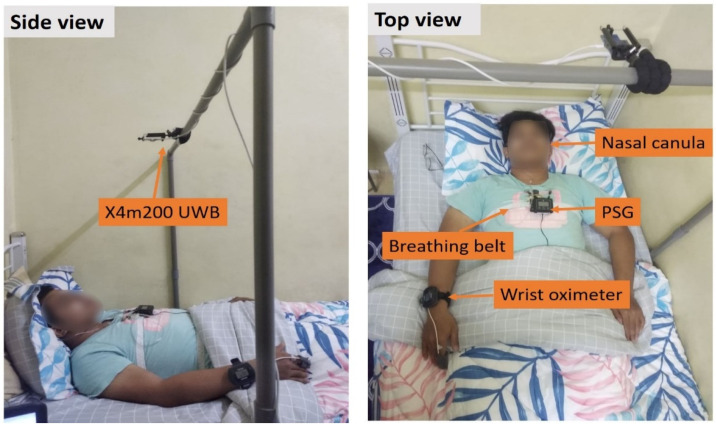
Experimental design for data collection.

**Figure 2 sensors-22-05249-f002:**
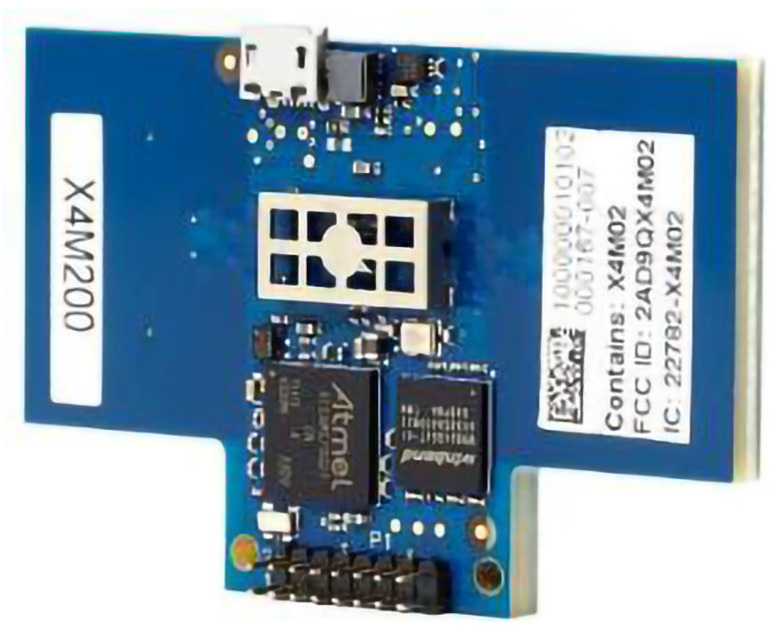
XeThru X4M200 UWB RADAR sensor.

**Figure 3 sensors-22-05249-f003:**
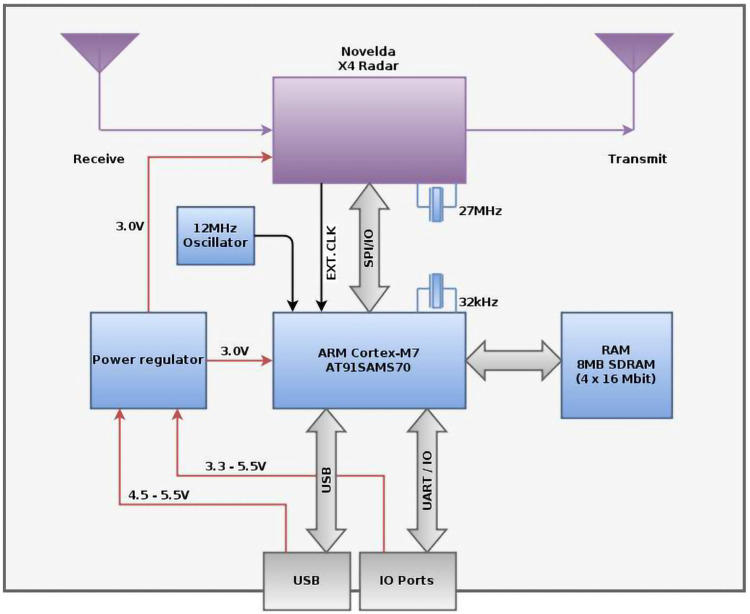
Block diagram of UWB X4M200 RADAR sensor. Retrieved 30 June 2022, from: https://novelda.com.

**Figure 4 sensors-22-05249-f004:**
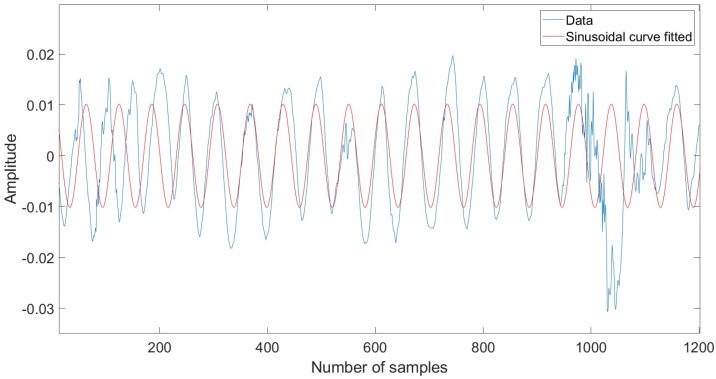
Sample of signal with sine curve fit above threshold value.

**Figure 5 sensors-22-05249-f005:**
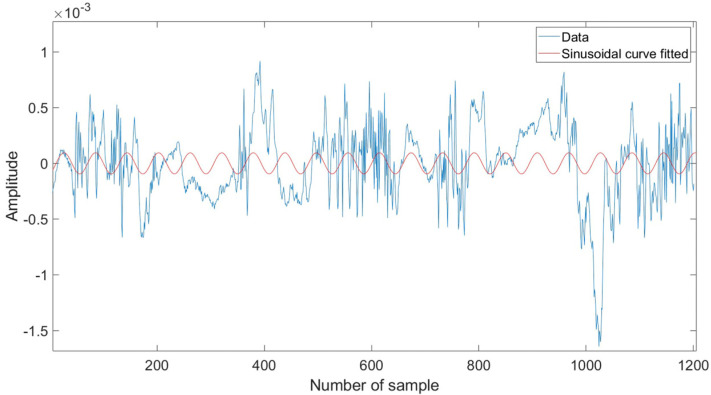
Sample of signal with sine curve fit below threshold value.

**Figure 6 sensors-22-05249-f006:**
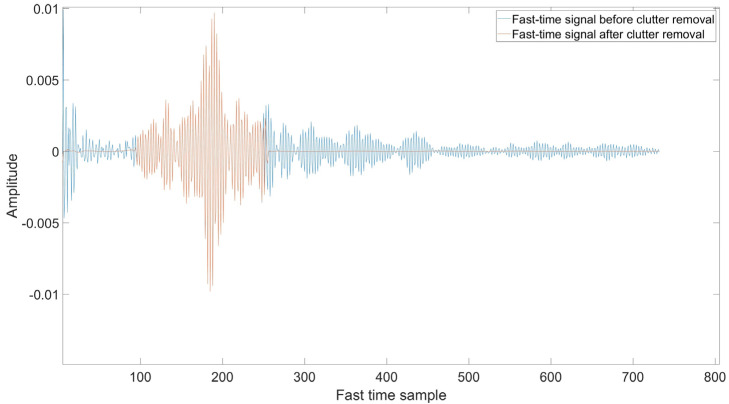
Clutter removal signal.

**Figure 7 sensors-22-05249-f007:**
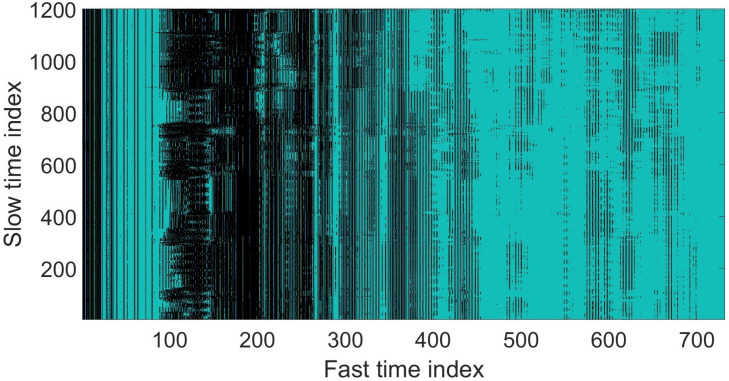
Fast time signal before clutter removal.

**Figure 8 sensors-22-05249-f008:**
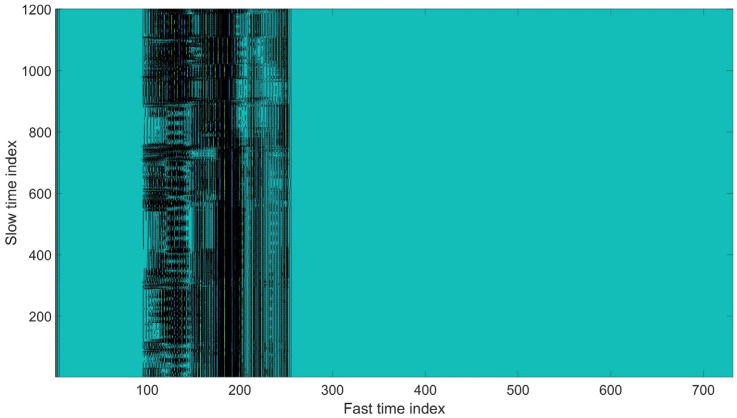
Fast time signal after clutter removal.

**Figure 9 sensors-22-05249-f009:**
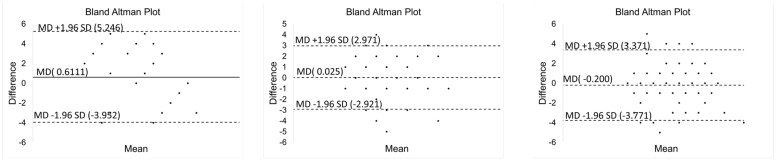
Bland Altman Plot for SBDA with PSG data for all 15 subjects.

**Table 1 sensors-22-05249-t001:** Subjects detail.

Subjects	Age	Weight (kg)	Height (cm)	Neck Circumference (inch)	Gender	Epworth Score	STOPBANG Score	AHI	Rating
1	27	65	163	12.5	Female	9	1	4	Normal
2	26	57	163	12.0	Female	7	1	3.7	Normal
3	28	53	151	13.5	Female	6	0	2.8	Normal
4	28	40	158	12.5	Female	9	1	0.8	Normal
5	23	50	170	12.0	Female	8	1	0.4	Normal
6	42	100	162	16.5	Female	5	2	12	Abnormal
7	36	54	159	14.5	Female	6	0	1.3	Normal
8	24	76	150	15.0	Female	6	1	2.6	Normal
9	41	68	163	13.0	Female	13	1	5.4	Abnormal
10	42	101	169	17.0	Male	3	6	23.2	Abnormal
11	21	75	175	15.3	Male	8	2	0.8	Normal
12	42	108	168	15.0	Male	19	4	24.5	Abnormal
13	27	48	167	13.0	Male	9	2	5.5	Abnormal
14	38	81	178	15.2	Male	4	3	18.1	Abnormal
15	25	67	165	14.0	Male	8	3	2.7	Normal

**Table 2 sensors-22-05249-t002:** Calculated energy ratio for 15 subjects.

Subject	IMF 1–IMF **3**	IMF **4**	IMF **5**	IMF **6**	IMF **7**	IMF 8–IMF **10**
1	-	0.0241	0.6642	0.5175	0.4550	-
2	-	0.1725	0.7565	0.6546	0.4040	-
3	-	0.0495	0.6554	0.5013	0.6563	-
4	-	0.2752	0.6701	0.5753	0.0051	-
5	-	0.1694	0.7490	0.5828	0.2768	-
6	-	0.3934	0.8145	0.6163	0.6916	-
7	-	0.5834	0.6768	0.5548	0.2016	-
8	-	0.0545	0.7918	0.5246	0.3457	-
9	-	0.2868	0.6443	0.6721	0.1725	-
10	-	0.5909	0.7223	0.5038	0.0341	-
11	-	0.0836	0.7718	0.6402	0.0020	-
12	-	0.0728	0.6104	0.5225	0.1039	-
13	-	0.2947	0.6174	0.6390	0.2253	-
14	-	0.1167	0.6145	0.5879	0.1146	-
15	-	0.2936	0.6250	0.5956	0.4659	-

**Table 3 sensors-22-05249-t003:** Summary of percentage error.

Percentage Error			
Subject	Autocorrelation + FFT (%) [45]	Mean Subtraction + FFT (%) [46]	SBDA (%)
1	14.20	20.38	8.02
2	11.22	15.75	6.69
3	9.53	11.50	7.56
4	8.02	8.39	7.66
5	12.57	16.65	8.49
6	10.65	14.04	7.25
7	11.73	18.65	4.81
8	16.77	26.47	7.08
9	7.11	8.20	6.02
10	9.91	15.77	4.05
11	9.56	13.24	5.88
12	9.98	14.96	5.01
13	9.81	14.89	4.73
14	9.64	14.83	4.45
15	9.46	14.76	4.17

## Data Availability

Not applicable.

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
