# Peer review of "Non-Contact Breathing Monitoring Using Sleep Breathing Detection Algorithm (SBDA) Based on UWB Radar Sensors"

_sensors, 2022, doi:10.3390/s22145249_

Round 1

Reviewer 1 Report

The paper investigates a Sleep Breathing Detection Algorithm (SBDA) for sleep breathing monitoring with an ultra-wideband radar. While the described topic is of interest for the community, some comments should be addressed before publication.

1)      The authors propose to use a method based on a UWB radar for the BR sleep monitoring. While it is clear why UWB radars have several advantages compared to Doppler ones, why did not you consider FMCW ones? Several works in the literature show how this radar topology can be used for the proposed application.

2)      The authors state that, during the measurements, the subjects lay on the bed and slept in their preferred position. Since the different targets were likely in different positions and had different orientations towards the antenna, how is possible to directly compare the results? What is the amplitude of the considered body movements?

3)      Would it be possible to use the proposed algorithm for heart rate detection? How would this technique compare to the ones proposed in the literature?

Reviewer 2 Report

In this paper, the authors aim to extract the respiration rate while sleeping. Although the topic is interesting and suitable for this journal, in my opinion is not suitable for publication.

There are few and not clearly reported radar measurements. The radar working mode and the breathing measurement steps are not detailed. I reported additional details hereafter.

This statement: “UWB radar has several benefits over doppler radar, such as inexpensive, high resolution, resistance to luminance or darkness, and good materials penetration performance [15].” Is not correct. Indeed, these are also advantages of Doppler or FMCW radars. You can read additional information in the following references: 10.1109/JSEN.2019.2941198, 10.1109/METROI4.2019.8792905. “It comprises a UWB radar supported by a Polyvinyl chloride (PVC) bar having an approximate range of 0.5 to 2 meters from the human chest to prevent the radar from moving or shaking”: Taking into account also the radar motion is another interesting point, in the literature there are different examples concerning vital sign detection in presence of radar self-motion.

It is not clear how you remove the clutter and the random body motion. The random body motion is not easily predictable, thus a simple fitting might be not appropriate. I am dubious that this method will work and will be applicable to different scenarios. The clutter remotion shown in Figs. 3,4, looks like a simple range gating.

In the comparison section, you should consider also different radar working modes, like Doppler and FMCW, to make the comparison more general.

J. E. Kiriazi, S. M. M. Islam, O. Borić-Lubecke and V. M. Lubecke, "Sleep Posture Recognition With a Dual-Frequency Cardiopulmonary Doppler Radar," in IEEE Access, vol. 9, pp. 36181-36194, 2021, doi: 10.1109/ACCESS.2021.3062385.

I suggest to expand the theoretical section, to detail the processing steps, particularly on the radar side and clarify what are the benefits from the radar point of view. Without these additional details, the procedure would be hardly reproduceable.

Round 2

Reviewer 1 Report

The authors answered almost all the reviewer's comment except for the one about the effect of body position/movements.

While it has been proven that it is possible to measure the vital signs independently of the body orientation, the presence of movements during the measurements is still an unsolved issue. If the authors did something to this extent they should clarify it in the text. Concerning the subject positions during the measurement, this is an element that should be clarified in the adopted measurement protocol to make the results comparable.

Reviewer 2 Report

I thank the authors for the effort put in revising the document.
However, there are still some concerns to be addressed.

1.Your answer: “Thank you for the comment. We have read the following references. The statement has been deleted from the manuscript.”. Of course, it is not enough that you read the references because it is important that also the readers will have the possibility to read the same references. Therefore, all the mentioned references should be added in the text.

2. Although your work is not focused on RSM, you should mention the problem. The breathing signal is very tiny, thus it is not it is not sure that the PVC bar will be completely stationary or other researchers may try to reproduce your setup with materials subject to small vibrations.

3. I am not convinced that your method will work under slightly different conditions. The human subject could perform periodic movements (like the typical body motion when the subject listens music) that might be confused with the breathing signal.

4. A comparison with different radars, like Doppler or FMCW is instead very interesting, also if an accurate comparison cannot be proposed due to the different techniques. Indeed, the reader should understand the advantages and disadvantages of using the UBW technology compared to the other techniques.

Round 3

Reviewer 1 Report

All the reviewer's comments have been addressed.

Reviewer 2 Report

The authors addressed all my concerns